# An integrated multi-omics analysis of sleep-disordered breathing traits implicates P2XR4 purinergic signaling

Nuzulul Kurniansyah[1], Danielle A. Wallace [1], Ying Zhang[1], Bing Yu[2], Brian Cade [1,3,4], Heming Wang [1,3,4], Heather M. Ochs-Balcom [5], Alexander P. Reiner [5,6], Alberto R. Ramos[7], Joshua D. Smith[8], Jianwen Cai [9], Martha Daviglus[10], Phyllis C. Zee[11], Robert Kaplan[12,13], Charles Kooperberg [12], Stephen S. Rich [14], Jerome I. Rotter[15], Sina A. Gharib[16], Susan Redline [1,4] & Tamar Sofer [1,4,17 ✉]

Sleep Disordered Breathing (SDB) is a common disease associated with increased risk for cardiometabolic, cardiovascular, and cognitive diseases. How SDB affects the molecular environment is still poorly understood. We study the association of three SDB measures with gene expression measured using RNA-seq in multiple blood tissues from the Multi-Ethnic Study of Atherosclerosis. We develop genetic instrumental variables for the associated transcripts as polygenic risk scores (tPRS), then generalize and validate the tPRS in the Women's Health Initiative. We measure the associations of the validated tPRS with SDB and serum metabolites in Hispanic Community Health Study/Study of Latinos. Here we find differential gene expression by blood cell type in relation to SDB traits and link *P2XR4* expression to average oxyhemoglobin saturation during sleep and butyrylcarnitine (C4) levels. These findings can be used to develop interventions to alleviate the effect of SDB on the human molecular environment.

[1] Division of Sleep and Circadian Disorders, Brigham and Women's Hospital, Boston, MA, USA. [2] Department of Epidemiology, Human Genetics, and Environmental Sciences, School of Public Health, The University of Texas Health Science Center at Houston, Houston, TX 77030, USA. [3] The Broad Institute of MIT and Harvard, Cambridge, MA, USA. [4] Division of Sleep Medicine, Harvard Medical School, Boston, MA, USA. [5] Department of Epidemiology and Environmental Health, School of Public Health and Health Professions, University at Buffalo, The State University of New York, Buffalo, NY, USA. [6] Department of Epidemiology, University of Washington, Seattle, WA, USA. [7] Department of Neurology, University of Miami Miller School of Medicine, Miami, FL, USA. [8] Northwest Genomic Center, University of Washington, Seattle, WA, USA. [9] Department of Biostatistics, University of North Carolina, at Chapel Hill, NC, USA. [10] Institute for Minority Health Research, University of Illinois at Chicago, Chicago, IL, USA. [11] Division of Sleep Medicine, Department of Neurology, Northwestern University, Chicago, IL, USA. [12] Division of Public Health Sciences, Fred Hutchinson Cancer Center, Seattle, WA, USA. [13] Department of Epidemiology & Population Health, Department of Pediatrics, Albert Einstein College of Medicine, Bronx, NY, USA. [14] Center for Public Health Genomics, University of Virginia School of Medicine, Charlottesville, VA, USA. [15] The Institute for Translational Genomics and Population Sciences, Department of Pediatrics, The Lundquist Institute for Biomedical Innovation at Harbor-UCLA Medical Center, Torrance, CA, USA. [16] Computational Medicine Core, Center for Lung Biology, UW Medicine Sleep Center, Department of Medicine, University of Washington, Seattle, WA, USA. [17] Departments of Medicine and of Biostatistics, Harvard University, Boston, MA, USA. ✉email: tsofer@bwh.harvard.edu

Sleep-disordered breathing (SDB) is a common disorder, affecting an estimated 24% of male and 9% of female adults in the U.S.[1]. SDB is characterized by episodic periods of breathing cessations and reductions during sleep, often accompanied by oxyhemoglobin desaturation[2,3], and is associated with cardiometabolic, vascular, and cognitive outcomes[4–7]. SDB is also strongly associated with inflammation[8,9]. While obesity is a strong risk factor for SDB, SDB is also heritable independent of body mass index (BMI)[10,11]. The underlying molecular processes by which SDB affects health outcomes are still being studied[12], with interest in understanding the effect of SDB-related hypoxia during sleep on cardiometabolic and vascular measures in humans and in animal models[13–15].

In investigating the molecular changes caused by SDB, previous studies showed changes in distributions and activation of white blood cells[16–18] and inflammatory cytokines[19] in individuals with obstructive sleep apnea (OSA). Other studies reported changes in gene expression in white blood cells following treatment using continuous positive airway pressure (CPAP), or following CPAP withdrawal[20–23], supporting a causal role between SDB-related physiological stressors (such as hypoxia) and immune cell gene expression. Some studies, including those from our group, also reported cross-sectional transcriptomic associations with SDB measures from observational studies[20,24]. However, these studies focused on a single cell population, and it is unknown whether and how transcriptional effects of SDB differ among circulating leukocyte subpopulations. Likewise, it is yet unknown how SDB-alterations in gene expression translate to metabolic changes. A few previous studies reported associations of blood metabolites with SDB phenotypes, independently of transcriptomics. For example, one study reported change in serum metabolite levels, evaluated on an untargeted platform that surveyed a few hundred metabolites, after multi-level sleep surgery[25]. Most other studies considered specific, targeted metabolite changes in sleep disorders (see reviews in ref. [26]).

Large, untargeted, omics surveys are now becoming available in cohort studies, providing an opportunity to study the association of SDB with well-defined, genetically-regulated molecular measures. We deploy a systems biology approach integrating genomic, transcriptomic, and metabolomic data to identify potential pathways in tissue-specific mechanisms driving SDB-related morbidity.

Utilizing data from the Multi-Ethnic Study of Atherosclerosis (MESA) and the Hispanic Community Health Study/Study of Latinos (HCHS/SOL), we examined multi-omics data to investigate signaling mechanisms underlying SDB traits. First, we used transcriptomics data measured in peripheral blood mononuclear cells (PBMCs), T-cells and monocytes, assayed by the Trans-Omics for Precision Medicine (TOPMed) program, to perform transcriptome-wide association study of SDB-related phenotypes (measured via overnight polysomnography) in MESA. We compared the results across different peripheral blood cell populations. With these data, we constructed transcript polygenic risk scores (tPRS) predicting transcript expression using genetic data. Next, we built these tPRS in the Women's Health Initiative (WHI) and tested them for association and generalization with their transcripts in whole blood. We calculated the tPRS that generalized in HCHS/SOL. Finally, we applied these tPRS to SDB traits and metabolites in HCHS/SOL to investigate how SDB phenotypes potentially propagate via transcription to metabolic changes in serum, and on the other direction, to assess potential reverse association by which transcript expression causes changes in SDB phenotypes.

Utilizing polysomnography, genetic, RNA-seq, and metabolomic data from multiple independent cohorts, we identified gene transcript expression patterns associated with leukocyte cell populations, SDB traits (AHI, blood oxygen levels), and blood metabolites. Genes represented by the transcripts associated with

SDB traits were related to hypoxia, neurotransmission, and thrombolytic activity. One main finding of a complete "chain" of association between a validated tPRS, a metabolite, and an SDB trait included the tPRS for *P2XR4*, a gene that encodes a purinergic receptor. Higher expression of *P2XR4* was associated with lower average oxyhemoglobin saturation during sleep and higher butyrylcarnitine, an indicator of fatty acid metabolism. SDB traits and obstructive sleep apnea are risk factors for the development of cardiovascular disease, and prior research has reported associations between *P2XR4* expression levels and cardiovascular function[27,28], as well as butyrylcarnitine and heart failure[29]. Therefore, our results suggest a mechanistic pathway for the role of purinergic signaling in SDB, which may have implications for the development of cardiovascular disease.

## Results

**Sample characteristics**. Characteristics of the MESA population that participated in the TOPMed omics study, the sleep study, and the smaller T-cells and monocytes analyses are provided in Supplementary Data 1; characteristics of the HCHS/SOL participants with genetic and metabolite data are provided in Supplementary Data 2. MESA individuals are a multi-ethnic sample, 69 years old on average during MESA exam 5, and 52% female. HCHS/SOL individuals are from diverse Hispanic/Latino backgrounds with a mean age of 46 years, and 59% female. SDB phenotypes were more severe in MESA, with average AHI = 18.6, MinO2 = 83, and AvgO2 = 94.1, in contrast to HCHS/SOL with average AHI = 6.4, MinO2 = 87.1, and AvgO2 = 96.4, consistent with the older age of the MESA sample. Characteristics of the WHI participants with RNA-seq data used to validate the transcript PRS are provided in Supplementary Data 3. WHI individuals are from a multi-race and ethnic sample and are 80 years old on average at the Long-Life Study exam when RNA was extracted and are all females.

**SDB phenotypes for oxyhemoglobin saturation and AHI are linked to tissue-specific changes in the transcriptome**. In MESA, we identified 96 and 24 differentially expressed transcripts (Supplementary Data 4–7) with FDR *p* value <0.1 in unadjusted and adjusted BMI analyses, respectively, in the different cell types. Table 1 reports the top differentially expressed transcripts (FDR *p* value <0.05). Three transcripts, *AJUBA* (Ajuba LIM Protein), *ZNF665* (Zinc Finger Protein 665), and *TMC3-AS1* (TMC3 Antisense RNA 1, a long non-coding RNA), are significantly associated with AvgO2 and AHI in both analyses, in the direction of reduced expression with worse SDB measures (higher AHI, lower AvgO2). Supplementary Data 4–7 further report results from secondary analyses adjusting all associations with FDR *p* value <0.1 in the primary analysis to cardiometabolic causal risk factors of OSA, including pulse pressure, type 2 diabetes, waist-to-hip ratio, hemoglobin A1c, as well as to alcohol use. Adjustments were performed for each phenotype separately, and jointly. Throughout, association effect estimates remained similar to those from the primary analysis.

To visualize gene expression and compare across SDB traits and cell types, log-fold change in expression of all SDB-associated transcripts (*n* = 96 transcripts FDR *p* < 0.1) was illustrated with a heatmap for BMI-unadjusted (Supplementary Fig. 1) and BMI-adjusted (Supplementary Fig. 2) analyses, clustered using hierarchical clustering based on the correlation between the log-fold estimates. These results illustrate concordant and discordant patterns of differential gene expression by cell type (PBMCs, monocytes, and T-cells) and SDB trait (AvgO2, MinO2, and AHI). There are a few striking differences in gene expression, particularly the increased expression of *FAM106A, TMC3-AS1*,

**Table 1 Top results from the tissue-specific transcriptome-wide gene expression analysis of SDB phenotypes (FDR $p < 0.05$) in MESA.**

| Gene | AdjLogFC | p value | FDR p value | SDB trait | Cell type |
|---|---|---|---|---|---|
| Unadjusted for BMI | | | | | |
| AJUBA | 0.104 | 2.59E−06 | 0.050 | AvgO2 | PBMCs |
| PDGFC | −0.016 | 2.59E−06 | 0.050 | MinO2 | PBMCs |
| SIAE | −0.011 | 1.26E−06 | 0.020 | MinO2 | Monocytes |
| EMP1 | −0.030 | 6.91E−06 | 0.050 | MinO2 | Monocytes |
| LHFPL2 | −0.014 | 9.42E−06 | 0.050 | MinO2 | Monocytes |
| ZNF665 | −0.011 | 2.74E−07 | 0.003 | AHI | T-cells |
| FAM106A | −0.032 | 7.69E−06 | 0.047 | AHI | T-cells |
| TMC3-AS1 | −0.022 | 2.74E−07 | 0.003 | AHI | T-cells |
| Adjusted for BMI | | | | | |
| AJUBA | 0.116 | 2.59E−07 | 0.005 | AvgO2 | PBMCs |
| ZNF665 | −0.012 | 5.49E−07 | 0.010 | AHI | T-cells |
| DUX4L27 | −0.029 | 6.59E−06 | 0.043 | AHI | T-cells |
| TMC3-AS1 | −0.021 | 7.14E−06 | 0.043 | AHI | T-cells |

The table provides significant results from analyses unadjusted ($n = 8$ transcripts) and adjusted ($n = 4$ transcripts) for BMI. Column names are in bold text. "AdjLogFC" is the covariate-adjusted log2-fold change in gene expression per 1 unit increase in SDB exposure. $p$ value is the empirical $p$ value (accounting for the distribution of $p$ values across 100 random permutations of the data), and FDR $p$ value is the $p$ value following FDR adjustment using the Benjamini–Hochberg procedure. For AvgO2 and MinO2, negative AdjLogFC indicates increased expression with worse SDB symptoms. For AHI, positive AdjLogFC indicates increased expression with worse SDB symptoms.

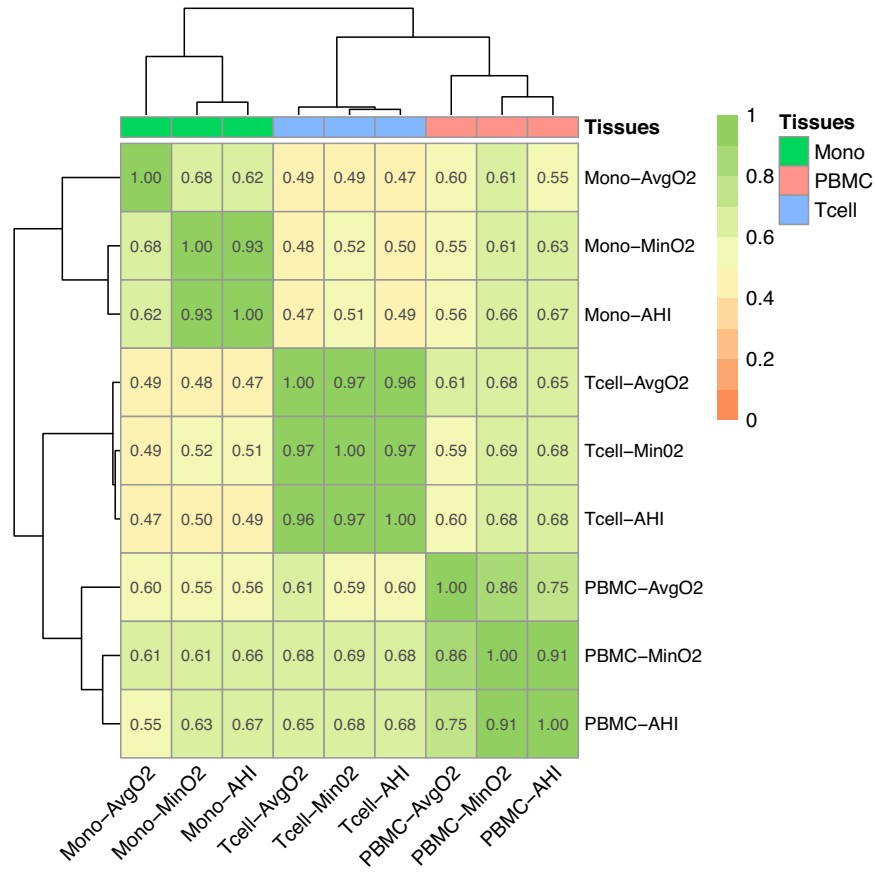

**Fig. 1 Spearman correlations between estimated log-fold changes in gene expression across SDB phenotypes and blood cell types without BMI adjustment in MESA.** Heatmap illustrating the Spearman correlations of log-fold change of transcript expression by tissue type (monocytes, T-cells, PBMCs) and SDB phenotype (AvgO2, MinO2, AHI) in MESA. Correlations were computed over genes with FDR $p < 0.1$. Color legend portrays Spearman $R^2$ (no/weak correlation = light yellow; complete/strong correlation = green). Estimated AHI effect sizes were flipped prior to computation of correlations so that they match the direction of MinO2 and AvgO2.

*SERPINE2*, *LA16c-312E8.4*, and *DUX4L27* in T-cells in association with better SDB measures (lower AHI, higher MinO2 and AvgO2) compared to monocytes and PBMCs in which the corresponding expressions tended to decrease, whereas expressions of *EMP1*, *SIAE*, *PDGFC*, and *LHFPL2* were decreased across

tissues in improved SDB phenotypes. To further investigate the overall patterns in gene expression in relation to tissue type and SDB traits, a heatmap of the Spearman correlation of the log-fold expression estimates of SDB phenotypes was plotted in Fig. 1. Within cell types, the SDB traits AHI and MinO2 had the highest

correlation for gene expression (Spearman $R^2$ between 0.91 and 0.97), whereas AvgO2 associations had lower correlations with AHI and MinO2 associations, especially in monocytes. A heatmap of correlations of estimated log-fold gene expression changes (FDR $p$ value <0.1) with SDB phenotypes across tissues from analyses adjusted for BMI is shown in Supplementary Fig. 3. The correlations between the SDB effect estimates for gene expression across cell types are different, and generally higher, from the phenotypic correlations between the SDB phenotypes, which are at the range of 0.53 to 0.73 Spearman $R^2$ (Supplementary Fig. 4). When computing correlations over all genes, estimated associations between AvgO2 with gene expression had low correlation with the other phenotypes (Supplementary Fig. 5). A similar pattern of decrease in the correlations between AvgO2 expression associations and other SDB trait associations are observed in BMI-adjusted analyses (Supplementary Fig. 6).

**Construction and validation of transcript PRS.** We constructed tPRS for gene expression in monocytes using a few methods, focusing on transcripts that were associated with SDB exposures in our analysis. The performance of constructed tPRS was evaluated against whole-blood gene expression levels in $n = 1269$ WHI participants. Supplementary Fig. 7 visualizes the results, demonstrating that tPRS constructed using the clump and threshold for genome-wide SNPs, including *trans*-eQTLs and tPRS focusing on *cis*-eQTLs have similar results, and the same generalization rate as that of the prediXcan-based tPRS. However, prediXcan tPRS had opposite direction of association with one of the transcripts in WHI, and, both *cis*-eQTLs based tPRS (prediXcan and clump and threshold) were not available for some transcripts due to lack of transcript-associated SNPs near the coding region. Thus, we moved forward with the genome-wide approach. Of the 96 tPRS (BMI-unadjusted analysis) and 24 tPRS (BMI-adjusted analysis) tested, 26 and 9 tPRS were associated ($p < 0.017$) with gene expression (Supplementary Data 8 and 9) in whole blood and considered validated as instrumental variables (IVs).

**Evidence of causal association between transcripts and SDB phenotypes.** We tested the association of the validated tPRS, constructed in a cell-specific manner, with SDB phenotypes in HCHS/SOL (Supplementary Data 10). Of the 26 tested in BMI-unadjusted analysis, 3 tPRS showed evidence of reverse association with SDB phenotypes ($p$ value <0.05), supporting a causal relationship between expression of these transcripts and SDB traits. Among them, the strongest association was of the tPRS for *P2RX4* (Purinergic Receptor *P2X 4*) in PBMCs in its association with AvgO2; one standard deviation (SD) increase in the tPRS was associated with a 1.9% decrease in AvgO2. Additionally, higher tPRS for *TUBB6* (Tubulin Beta 6 Class V) in monocytes was associated with lower MinO2. These directions of associations matched those observed in association analysis of the SDB phenotype and transcript expression in MESA. However, higher tPRS for *SEC14L2* (SEC14 Like Lipid Binding 2) in T-cells was associated with higher AHI, but this direction of association did not match that of the estimated AHI-transcript association in MESA. After BMI adjustment, only *P2RX4* in PBMCs tPRS remained associated with AvgO2 ($p$ value <0.05), as shown in Supplementary Data 11.

**Evidence of causal association between transcripts and metabolites.** We tested the relation between each validated tPRS and metabolites in HCHS/SOL. The tPRS for *P2RX4* and *CTD-2366F13.1* (also known as *MOCS2-DT*, MOCS2 Divergent

Transcript) were associated with a total of 6 and 7 metabolites in unadjusted BMI and adjusted BMI analyses (FDR $p$ value <0.05, Supplementary Data 12 and 13), respectively; the association "chains" are visualized in Fig. 2. Of the 7 metabolites, 4 of them (butyrylcarnitine, 1-stearoyl-2-arachidonoyl-GPE (18:0/20:4), linoleoyl-arachidonoyl-glycerol (18:2/20:4), and palmitoleoyl-linoleoyl-glycerol (16:1/18:2)) were also associated with AvgO2 (Supplementary Data 14). However, the AvgO2-metabolite associations did not remain statistically significant after BMI adjustment, suggesting that BMI, rather than SDB, may be driving these associations (Supplementary Data 15). Of the transcripts, *P2RX4* had evidence of a complete chain of association with SDB and metabolites ($p$ value <0.05) in the BMI-unadjusted analysis.

## Discussion

Here, we conducted a robust analysis of SDB phenotypes and their multi-omics correlates. We first identified transcriptome-wide tissue-specific changes in gene expression associated with sleep-related oxyhemoglobin saturation traits and AHI in MESA and then used those transcripts to develop genetic proxies for gene expression (tPRS). Next, we generalized and validated some of the tPRS in WHI. Finally, we utilized the validated tPRS to further study SDB phenotypes and metabolite associations in HCHS/SOL. Our results support SDB-related leukocyte alterations in gene expression and highlight signaling pathways related to inflammation, thrombosis, and neurotransmission.

SDB traits were associated with differential expression of many transcripts across three blood cell types (Supplementary Data 4 and 5), 96 genes with FDR $p$ value <0.1. Of the top transcripts (8 genes with FDR $p$ value <0.05), higher *AJUBA* expression was associated with higher AvgO2 and higher *PDGFC* expression was associated with lower MinO2 in PBMCs. *AJUBA* is a scaffold protein in the family of LIM domain-containing proteins, considered key regulators of the hypoxic response[30]. Recent research supports a role for *AJUBA* in interacting with retinoic acid receptor signaling in an in vitro model[31] and a role for indirectly limiting inflammation by maintaining mitochondrial quality control in a mouse model[32]; therefore, greater *AJUBA* expression may be associated with increased AvgO2 through pathways related to inflammation and retinoic acid. *PDGFC* encodes platelet derived growth factor C[33] and is upregulated during hypoxia in tumor cells[34]. The association between higher *PDGFC* expression and lower minimum oxygen saturation (MinO2) supports a role for *PDGFC* signaling in SDB-related hypoxia. In monocytes, increased expression of *EMP1*, *SIAE*, and *LHFPL2* was associated with lower MinO2, suggesting that expression of these genes increases as oxygen levels decrease. In line with these findings, prior studies have shown that *EMP1* expression increases during sleep loss and during hypoxia in cancer tissues[35,36]. While the functions of the esterase *SIAE* and the transmembrane protein *LHFPL2* are unclear, increased expression of each is a marker for poor cancer prognostic: higher expression of *SIAE* is linked to poorer prognosis for patients with multiple myeloma and higher expression of *LHFPL2* is linked to poorer prognosis for patients with liver cancer[37,38]. These cancer-related markers may be relating to SDB phenotypes due to the overlap between hypoxia and the hypoxic tumor microenvironment[39].

Correlations between leukocyte subsets and trait-specific gene expression (Fig. 1) supported an overall pattern of similarity between cell populations, but also highlighted some striking differences. For example, higher expression of the gene *HPCAL4* (Hippocalcin-Like Protein 4) is strongly associated with "worse" SDB phenotypes (higher AHI, lower AvgO2 and MinO2) in

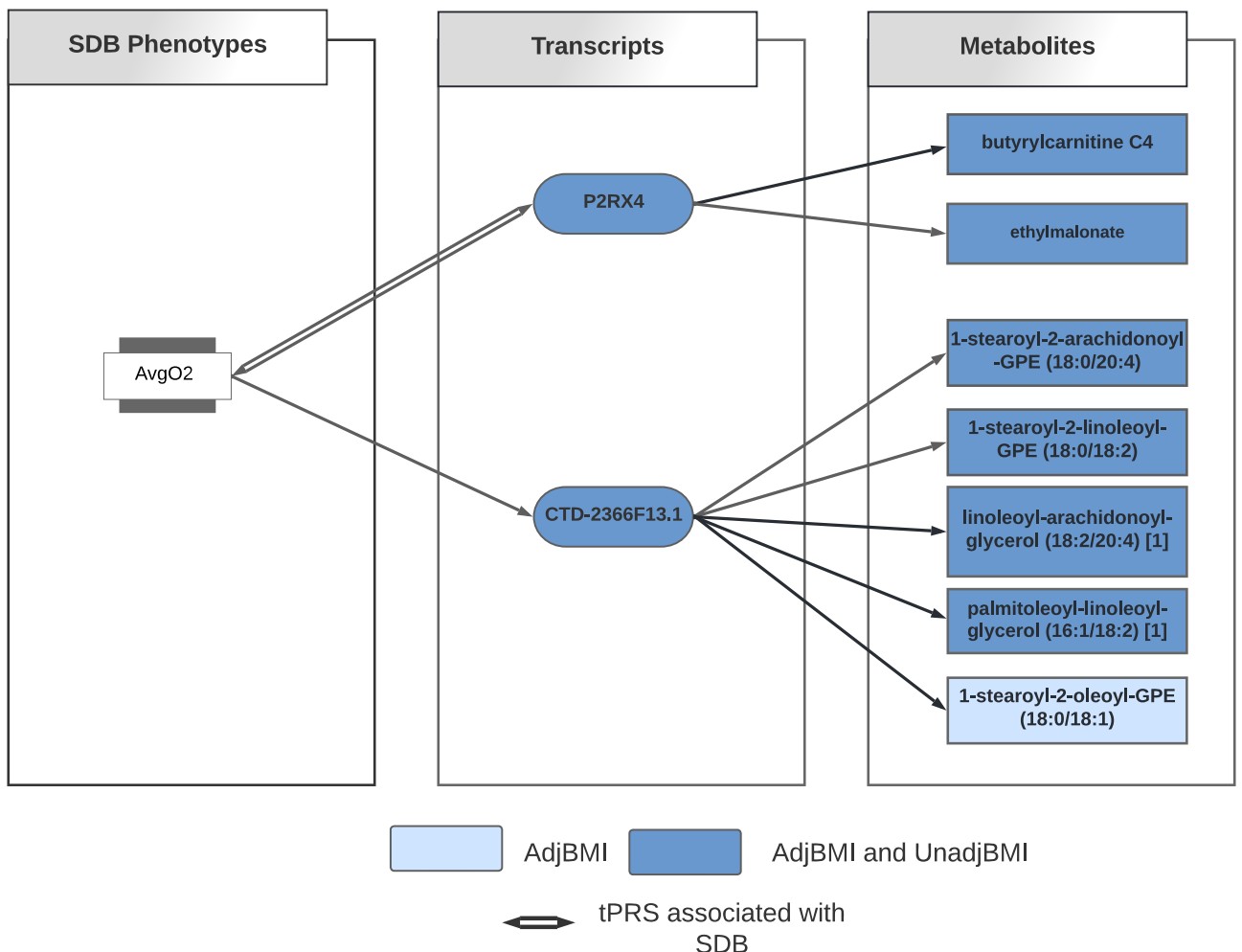

**Fig. 2 Identified association chains between AvgO₂, transcripts, and metabolites.** Diagram illustrating the "association chain" relationships between AvgO₂, tPRS, and metabolites in BMI-unadjusted and BMI-adjusted analyses. Dark blue color indicates association between tPRS and metabolites in both BMI-unadjusted and BMI-adjusted analyses; The light blue color indicates the association between tPRS and metabolite only in BMI-adjusted analysis.

monocytes, but this gene has weak associations with SDB phenotypes in T-cells and PBMCs. However, *FAM106A* and *SER-PINE2* have higher expression with worse SDB phenotypes in T-cells, but weak associations in PBMCs and monocytes. *SERPINE2* encodes glia-derived nexin (GDN, also referred to as protease nexin-1) with mixed evidence as a genetic factor for COPD[40–42]. *SERPINE2/GDN* inhibits the hypoxia-triggered serine protease thrombin[43,44], and a mouse model of *SERPINE2* deficiency causes excess thrombin activity and overproduction of cytokines in the lungs[45], suggesting a role for *SERPINE2* in airway inflammation. Since OSA may be associated with a procoagulant state featuring high thrombin levels[46,47], the hypoxemia associated with OSA may lead to increased thrombin levels, affecting *SERPINE2* expression in T-cells. Similar to *FAM106A* and *SERPINE2*, *TMC3-AS1* expression in T-cell increased as AHI increased; because *TMC3-AS1* encodes a lncRNA that may inhibit the anti-inflammatory cytokine IL-10[48], increased expression could result in higher IL-10 levels, possibly as a compensatory response to greater AHI.

Of the top differentially expressed genes in MESA whose tPRS was validated in an independent cohort (WHI), only *P2RX4* was found to have a complete "chain" of association with SDB and metabolites when tested in another independent study, HCHS/SOL. Higher tPRS for *P2RX4* was associated with lower AvgO2, both with and without adjustment for BMI, and as such it is a candidate contributor to oxyhemoglobin saturation in SDB. *P2RX4* encodes a purinergic receptor for ATP, P2X₄, which may play a role in the neuroprotective effects of hypoxic preconditioning[49,50]. *P2RX4* tPRS was further associated with higher concentrations of the metabolite butyrylcarnitine, an indicator of fatty acid metabolism previously linked to BMI[51]. Higher oxygen tension promotes increased ATP production[52], which may in turn promote increased *P2RX4* expression (P2X₄ as a purinergic ATP receptor) and increased butyrylcarnitine[53]. *P2RX4* can have beneficial or detrimental effects depending on context. A mouse model of genetically increased *P2XR4* expression led to enhanced cardiovascular function[27] and prior research supports a role for *P2XR4* in heart contractility[28], suggesting that *P2XR4* may impact AvgO2 levels via alteration of cardiac force. Likewise, butyrylcarnitine was found to be associated with time of cardiac isovolumetric relaxation and may be a marker of heart failure[29], further linking *P2XR4* to cardiac function. However, ethanol is an inhibitor of *P2RX4*, and *P2RX4* has been associated with alcohol intake. Because alcohol consumption is a risk factor for SDB and can promote airway collapse[54–56], in a secondary analysis we adjusted for self-reported alcohol consumption in the MESA RNA-seq analysis. The results did not substantially change, suggesting that alcohol use is not driving this association; however, residual confounding by alcohol use is still possible.

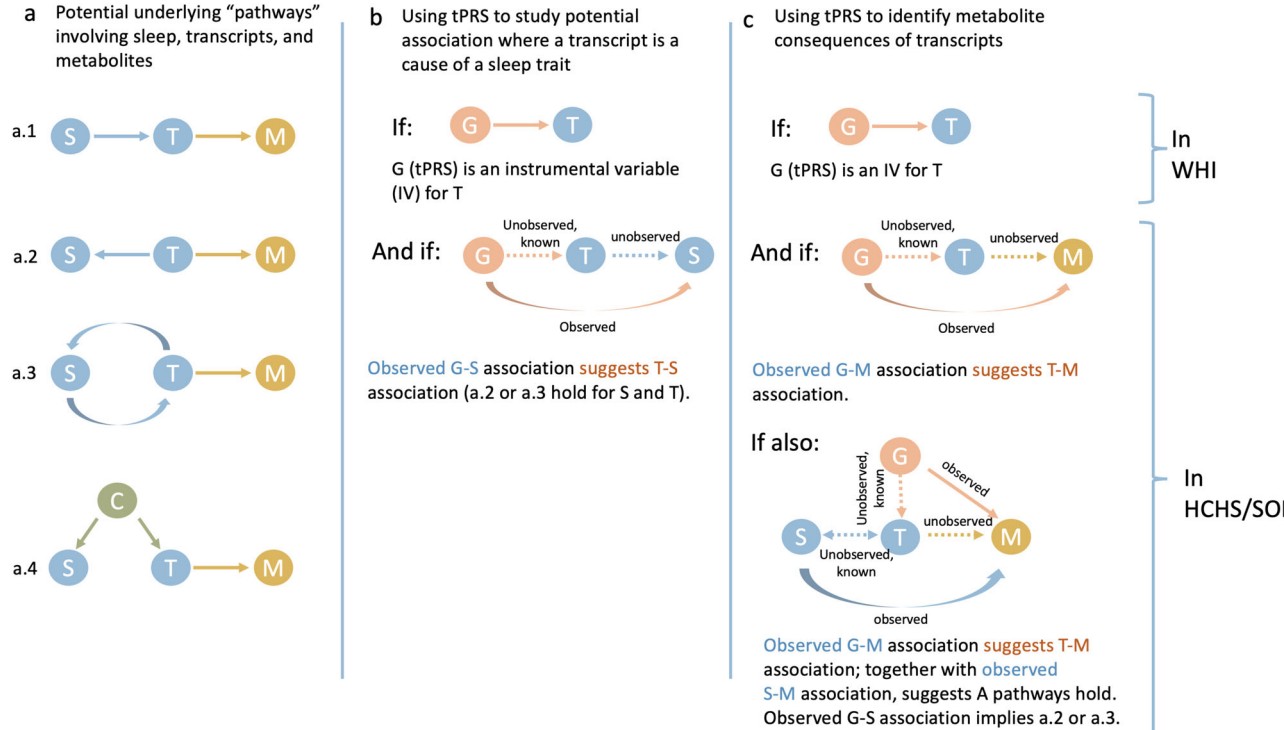

**Fig. 3 Potential association "chains" of SDB traits, transcript expression, and metabolites, addressed in this study.** The figure illustrates association chains, or pathways, potentially linking an SDB trait, a transcript, and a metabolite. Here we assume that an association between a sleep trait and a transcript was detected in MESA and is assumed "known" for follow-up analysis in WHI in HCHS/SOL. **a** demonstrates potential forms of causal associations between the sleep trait and the transcript, including (a.4) the settings where an association exists due to a common cause, e.g., BMI. Our metabolomics analysis may only detect transcript–metabolite associations, i.e., any sleep–metabolite link is via transcript levels. **b** demonstrates a potential conclusion from an association between a tPRS, validated in WHI and used as an instrumental variable (IV) of a transcript, and a sleep trait: if an association is detected, it provides evidence that changes in transcript levels are upstream (a cause of) changes in sleep trait levels. **c** demonstrates potential conclusions from analyses linking tPRS and a sleep trait to a metabolite. A tPRS is used to link a transcript to a metabolite, and an association, if exists, is likely causal. A sleep–metabolite association should exist if the sleep–transcript and tPRS–transcript associations hold, and therefore observing such an association validates the existence of any of A pathways. Further association between the tPRS and the sleep trait narrows down the potential association chains to a.2 or a.3.

While *CTD-2366F13.1* (*MOCS2-DT*) PBMCs tPRS was not associated with SDB traits in HCHS/SOL, levels of three of the four metabolites positively associated with *CTD-2366F13.1* tPRS were also negatively associated with AvgO2 (i.e., increased metabolite levels with reduced AvgO2): linoleoyl-arachidonoyl-glycerol (18:2/20:4), palmitoleoyl-linoleoyl-glycerol (16:1/18:2), and 1-stearoyl-2-arachidonoyl-GPE (18:0/20:4). The lack of observed association between the *CTD-2366F13.1* tPRS and AvgO2 suggests that it is likely that AvgO2 may cause expression changes in the gene, rather than the genetically-determined gene expression causes AvgO2 (Fig. 3). Therefore, lower AvgO2 may result in increased concentrations of these metabolites. linoleoyl-arachidonoyl-glycerol (18:2/20:4), has previously been positively associated with serum levels of the antioxidant alpha-tocopherol, also known as Vitamin E[57]. Levels of the component 2-Arachidonoylglycerol, an agonist of the CB1 and CB2 cannabinoid receptors, are increased in the brain during ischemia[58] and in macrophages in response to oxidative stress[59]. Palmitoleoyl-linoleoyl-glycerol (16:1/18:2), is a palmitoleic acid derivative that may be a marker of blood sugar regulation; it is also commonly used in baked goods[60]. Increased levels have previously been linked to in utero exposure to gestational diabetes[60]. Given the relationship between SDB and cardiometabolic disease, it is possible that these metabolites are associated with AvgO2 because of their links to the immune system and glycemic regulation. In fact, AvgO2 associations with these metabolites became null in a BMI-adjusted analysis.

There are several strengths and some limitations of our analysis. Our unique study design exploited a stepwise discovery/validation approach across multiple studies and optimized the availability of SDB-related datasets to study omics markers and SDB. First, we identified SDB-related transcripts. Next, we utilized genetic associations with gene expression to construct tPRS, serving as "genetic IVs": exposure variables that are likely associated with the gene transcripts and are specific to them, thus allowing for downstream association analysis and causal inference using these IVs instead of the transcript themselves[61]. The idea of using genetic variants as IVs is often used in Mendelian Randomization (MR) analysis. Our analysis is different than standard one-sample MR in that we did not estimate the effect of the transcript on the outcome, because we did not have access to RNA-seq in HCHS/SOL. However, for causality inference, it is sufficient to test the IV association with the outcome of interest[62]. We then studied the evidence for the effect of gene expression on SDB using the tPRS. Still, the exact form of association between the gene expression and SDB traits cannot be determined (Fig. 3). For example, if no tPRS–SDB association was detected, it is possible that this was due to lack of power. Even in the absence of tPRS–SDB association, the association between the SDB and tPRS can be due to either causal effect of SDB on tPRS, or confounding by a common cause of both. Notably, during the WHI validation step, many transcripts did not have significant tPRS associations and therefore were not carried forward for the genetic association analysis in HCHS/SOL; lack of validation may be due to cell type

differences, as we validated tPRS in WHI, where gene expression was measured in whole blood, unlike measurement in specific cell types in MESA. Finally, we leveraged the validated tPRS to test for associations of gene transcript expression with metabolites and connect possible "chains" of associations. All included cohorts are large and represent diverse populations in the U.S. Our sleep cohorts, HCHS/SOL and MESA, have objective sleep phenotype measurement without prior selection of participants based on specific phenotypes. Other limitations of our study include high multiple testing burden, performing procedures with multiple steps, utilizing multiple data in constructing tPRS, and differences in sample timing between blood sample collection and overnight PSG in MESA. However, genetic data should not be affected by differences in timing, and chronic conditions like SDB may be stable over time, making this limitation less of a concern. It is notable that the three blood cell types used in MESA are not distinct: PBMCs include monocytes and T-cells. Further, both monocytes and T-cells are also composed of more granular cell types. Statistical analyses within one cell type are generally powered to detect associations that hold across the component, more granular, cell types, and some cell type-specific associations may be masked. Overall, we utilize robust statistical methods and objective measures, integrating across multiple layers of biological measures, to interrogate the mechanisms driving SDB-related morbidity.

In summary, we examined multiple levels of biological information to investigate signaling mechanisms underlying SDB traits to better understand drivers of morbidity in SDB. Our results highlight differential gene expression by circulating leukocyte populations in relation to multiple SDB traits related to hypoxia, neurotransmission, and thrombolytic activity. Analyses with validated tPRS in independent cohorts support a mechanistic role for *P2XR4* purinergic signaling in SDB, a gene known to influence cardiac function, which is relevant to SDB as both a risk factor and an outcome. While further research is necessary to confirm these findings, they suggest that *P2XR4* signaling may alter oxygen levels during sleep. In the future, we hope that more data will become available with more granular cell-specific transcriptomics, to better understand cell-specific responses involved in SDB, as well as their validation. Large complementary genetic datasets with unbiased genetic associations of SDB, gene transcripts, and metabolomics, will further facilitate causal inference via Mendelian randomization analysis. Overall, we applied robust methods to integrate multi-omics data and SDB data to discover mechanisms underlying multiple SDB traits. Our multi-dimensional approach using large population cohorts is a promising approach to unravel biological underpinnings of complex human disorders.

## Methods

**Overall study design and purpose**. The overall purpose of the study was to investigate the multi-omics signaling mechanisms underlying SDB traits to better understand possible drivers of morbidity in SDB. The study design and purpose of each analysis component is illustrated in Fig. 4. Briefly, panel a demonstrates the set of associations investigated: SDB phenotypes lead to transcriptional changes which in turn lead to metabolic changes; panel b describes the analysis steps taken to study the potential chain of associations, and the goal of each of these steps. To optimize the available sample and leverage the fact that transcription is, to some extent, genetically determined, we utilized two separate cohorts to identify the biological components associated sleep exposure to metabolomic changes. Figure 3 further illustrates potential causal relationships underlying a set of measures, and the assumptions that we used to interrogate some of them. Thus, we first performed transcriptome-wide association studies for SDB phenotypes in MESA. For each transcript associated with a SDB trait (FDR *p* value <0.1), we used genetic data to construct a transcript Polygenic Risk Score (tPRS) to serve as a predictor of the transcript. Next, to reduce false positive associations in subsequent analyses, we constructed these tPRS in the WHI and tested their association and generalization with their transcripts in whole blood. We proceeded with tPRS results that generalized (*p*

value <0.05), and constructed and tested them latter for association with SDB phenotypes in HCHS/SOL. If a tPRS was associated with the SDB phenotype in HCHS/SOL, it was interpreted as evidence of reverse association, i.e., the transcript may contribute to SDB. We then calculated the association of the tPRS with metabolites in HCHS/SOL. Lastly, we used another analytic step to support the existence of an association chain linking a sleep exposure, a transcript, and a metabolite: we required evidence of association between the sleep exposure and the metabolite in HCHS/SOL (i.e., any of the potential pathways in Fig. 3a). If the tPRS was associated with the sleep exposure in HCHS/SOL, it lent support to association chains where the transcript affects both SDB and metabolite levels.

**Participating studies**. As described in Fig. 1, our analysis included three studies: MESA, WHI, and HCHS/SOL each contributing to different analytical steps. The three studies are described in the Supplementary Note. In brief, MESA, our primary study used for discovery of SDB–transcript associations, is a longitudinal cohort study[63]. The first and fifth MESA exams took place between 2000–2002, and 2010–2012, respectively, and whole blood was drawn from participants in both exams. For about 1000 participants, blood was used later for RNA extraction in at least one of the exams. In addition, a sleep study ancillary to MESA occurred shortly after MESA exam 5 during 2010–2013. Sleep study participants underwent single night in-home polysomnography[64]. The number of individuals with each type of data and at each time point (exam 1 and exam 5) varies. Supplementary Fig. 8 visualizes the data flow and overlaps across the various measures used in this study: whole-genome genotyping, RNA-seq, and sleep. All MESA participants provided written informed consent, and the study was approved by the Institutional Review Boards at The Lundquist Institute (formerly Los Angeles BioMedical Research Institute) at Harbor-UCLA Medical Center, University of Washington, Wake Forest School of Medicine, Northwestern University, University of Minnesota, Columbia University, and Johns Hopkins University.

The WHI was here used to identify tPRS that could be confidently used as IVs for their traits. It is a prospective national health study focused on identifying optimal strategies for preventing chronic diseases that are the major causes of death and disability in postmenopausal women[65]. In all, 11,071 WHI participants have whole-genome sequencing data via TOPMed, and 1274 of these participants have RNA-seq measured in venous blood via TOPMed. All WHI participants provided informed consent and the study was approved by the IRB of the Fred Hutchinson Cancer Research Center.

The HCHS/SOL was used to establish association chains that include an SDB trait, a transcript, and a metabolite. It is a longitudinal cohort study of U.S. Hispanics/Latinos[66,67]. The HCHS/SOL baseline exam occurred on 2008–2011, where 16,415 participants were enrolled. HCHS/SOL individuals who consented further participated in an in-home sleep study, using a validated type 3 home sleep apnea test recording airflow (via nasal pressure), oximetry, position, and snoring. Genetic data were measured and imputed to the TOPMed freeze 5b reference panel, for individuals who consented at baseline[68,69]. Metabolomic data were also measured for *n* = ~4000 individuals selected at random out of those with genetic data[70]. Supplementary Fig. 9 provides the data flow in HCHS/SOL, focusing on individuals with genetic data and wide consent for genetic data sharing. The HCHS/SOL was approved by the institutional review boards (IRBs) at each field center, where all participants gave written informed consent, and by the Non-Biomedical IRB at the University of North Carolina at Chapel Hill, to the HCHS/SOL Data Coordinating Center. All IRBs approving the study are: Non-Biomedical IRB at the University of North Carolina at Chapel Hill. Chapel Hill, NC; Einstein IRB at the Albert Einstein College of Medicine of Yeshiva University. Bronx, NY; IRB at Office for the Protection of Research Subjects (OPRS), University of Illinois at Chicago. Chicago, IL; Human Subject Research Office, University of Miami. Miami, FL; Institutional Review Board of San Diego State University, San Diego, CA.

**RNA sequencing**. For both MESA and WHI, RNA-seq was performed via the Trans-Omics in Precision Medicine (TOPMed) program. In MESA, RNA-seq was generated from three blood cell types: peripheral blood mononuclear cells (PBMCs; ~*n* = 1200 measured in blood from visits 1 and 5), and specific components: T-cells, and monocytes (referred to as T-cell and Mono, for both: *n* = 416), measured in blood from visit 5. Samples were sequenced at the Broad Institute and at the North West Genomics Center (NWGC). Both centers used harmonized protocols. RNA samples quality was assessed using RNA Integrity Number (RIN, Agilent Bioanalyzer) prior to shipment to sequencing centers. QC was re-performed at sequencing centers by RIN analysis at the NWGC and by RNA Quality Score analysis (RQS, Caliper) at the Broad Institute. A minimum of 250 ng RNA sample was required as input for library construction, performed using the Illumina TruSeq[TM] Stranded mRNA Sample Preparation Kit. RNA was sequenced as 2x101bp paired-end reads on the Illumina HiSeq 4000 according to the manufacturer's protocols. Target coverage was of ≥40 M reads. Comprehensive information about the RNA-seq pipeline used for TOPMed can be found in https://github.com/broadinstitute/gtex-pipeline/blob/master/TOPMed_RNAseq_pipeline.md under MESA RNA-seq pilot commit 725a2bc. Here we used gene-level expected counts quantified using RSEM v1.3.0[71]. RNA sequencing for WHI (whole

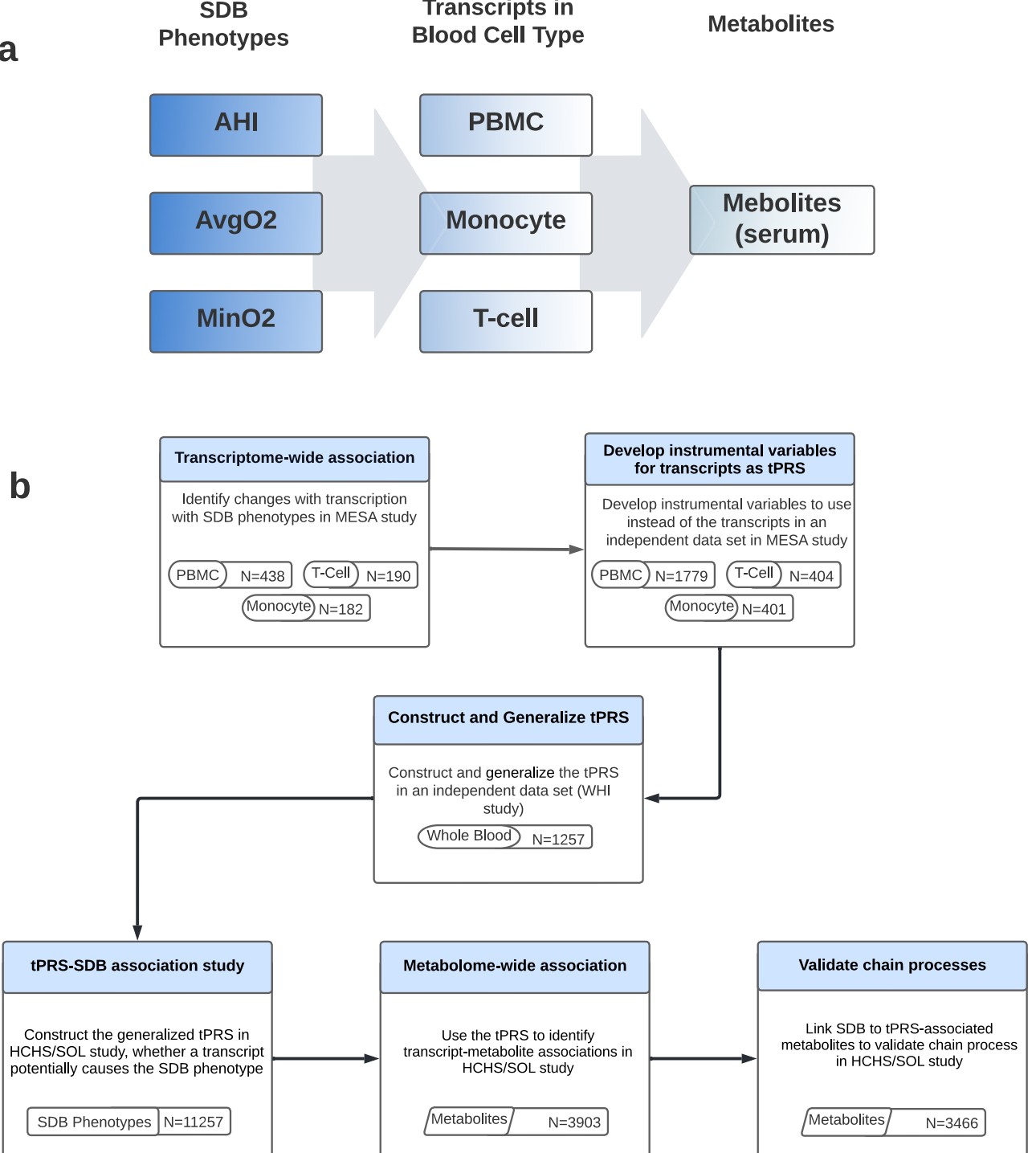

**Fig. 4 Overall study design of the reported analysis.** Flow charts illustrating the methodology and purpose of the analysis. The chart portrays the three SDB phenotypes evaluated in the analysis, the three blood tissues with transcript expression measurement, and demonstrates each step of the analysis with associated goals, cohorts, and reasons why step was performed. **a** Conceptual linking between SDB measures, transcript expression, and metabolites. **b** Analytic steps supporting the study of the conceptual links. PBMC tPRS analysis sample sizes in MESA correspond to data from two visits (some individuals were used twice, appropriately accounted for by mixed models).

blood) was performed at the Broad Institute using the unified TOPMed protocols. More information about RNA-seq in WHI is provided in the Supplementary Note.

unknown (unidentified) metabolites. Detailed methodologic information is provided elsewhere[70].

**Metabolomics data in HCHS/SOL**. Metabolomics profiling using fasting blood samples was conducted at Metabolon (Durham, NC) with Discovery HD4 platform in 2017. Serum metabolites were quantified with untargeted, liquid chromatography-mass spectrometry (LC-MS)-based quantification protocol[72,73]. The platform captured a total of 1136 metabolites, including 782 known and 354

**Phenotypic measures of sleep disordered breathing**. We used three SDB traits, as measured by overnight sleep studies in MESA and HCHS/SOL (methods above): (1) the Apnea-Hypopnea Index (AHI), defined in MESA as the number of apneas (breathing cessation) and hypopneas (at least 30% reduction of breath volume, accompanied by 3% or higher reduction of oxyhemoglobin saturation) per hour of

sleep, and in HCHS/SOL, due to differences in the recording montage compared to MESA, as the number of apnea or hypopnea events with 3% desaturation per hour of sleep; (2) minimum oxyhemoglobin saturation during sleep (MinO2), and (3) average oxyhemoglobin saturation during sleep (AvgO2).

**Testing the association between SDB and blood cell-specific transcriptome-wide gene expression**. We used the Olivia R package[74] to perform association analyses of gene expression in PBMCs, monocytes, and T-cells with each of the three SDB measures, separately and in a joint analysis in MESA. SDB phenotypes were treated as the exposures. In joint analysis, the three SDB phenotypes were modeled as three exposures in the same model. We followed the recommended Olivia pipeline. Briefly, we performed median normalization, and then filtered lowly expressed gene transcripts defined by removing transcripts with proportion of zero higher than 0.5, median value lower than 1, maximum expression range value lower than 5, and maximum expression value lower than 10. Transcript counts were log transformed after counts of zero were replaced with half the minimum of the observed transcript count in the sample. The analyses were adjusted for age, sex (validated using chromosomal checks), study center, race/ethnic group, and batch variables: plates, shipment batch, and study site. Because BMI is a strong risk factor for SDB and is assumed to be part of the causal chain, we conducted additional analyses adjusting for BMI. We computed empirical $p$ values to account for the highly skewed distribution of SDB phenotypes, which may lead to false negative associations if ignored. However, for joint SDB phenotypes analysis we used the multivariate Wald test, without further empirical $p$ values, because the permutation procedure only works when using a single exposure. Finally, we accounted for multiple testing by applying False-Discovery Rate (FDR) correction to each set of gene-based $p$ values corresponding to a single SDB phenotype, or the joint association, using the Benjamini–Hochberg (BH) procedure[75]. We carried forward transcript associations with FDR $p$ value <0.1 for additional analyses and visualized their association with SDB phenotypes via a hierarchically clustered heatmap.

In secondary analyses, we assessed the effect of adjusting for additional available phenotypes that prior work suggested as causal to SDB[76], and for alcohol use. Specifically, we applied the Olivia pipeline to estimate the same transcript-SDB phenotype associations using the same regression adjustment, while also including one of the additional risk factors as a covariate: pulse pressure, type 2 diabetes, waist-to-hip ratio, hemoglobin A1c, and alcohol use. An additional analysis included these five covariates together. We studied the association results for all transcripts with FDR $p$ value <0.1 in the main analyses once adjusting to these covariates. Description of these measures is provided in the Supplementary Note.

**Transcript polygenic risk scores (tPRS) construction and validation**. To develop tPRS, we first performed a genome-wide association study (GWAS) for each SDB-associated transcript using the MESA TOPMed WGS dataset; each GWAS adjusted for age (years), sex, study site, self-reported race/ethnic background, and 11 principal components, and analyses were restricted to genetic variants with a minor allele frequency of at least 0.05 (due to low sample size). For each GWAS, we used the fully adjusted two-stage procedure for rank-normalizing residuals in association analyses[77] to identify genetic variants associated with transcript expression. For PBMCs, we used transcript measures from the two MESA visits with RNA-seq data to increase power. To do this, we removed related individuals, and used a random effect model that accounted for individuals. Summary statistics from the GWAS for each transcript were used to develop PRS weights for the corresponding transcript. Next, we constructed tPRS in MESA. We applied clump and threshold methodology implemented in PRSice2 v2.3.1.e[78] using clumping parameters $R^2 = 0.1$, distance of 250 Kb, and three $p$ value thresholds: genome-wide significance ($5 \times 10^{-8}$), and levels of evidence considered "suggestive" ($10^{-7}$, $10^{-6}$). For each transcript, we constructed the three tPRS in WHI. A tPRS with the smallest $p$ value in association with the transcript in WHI, and also having $p$ value $<0.05/3 = 0.017$, was selected and considered validated. We also computed FDR-adjusted $p$ values based on all constructed tPRS (3 candidate tPRS per gene across all genes). To test the association of the tPRS with transcript in WHI, we used logistic mixed models, executed with the GENESIS R package[79] version 2.16.1. Each tPRS served as the exposure, and transcripts served as the outcome, here too using the two-stage procedure for rank-normalization[77]. Relatedness was modeled via a sparse kinship matrix among TOPMed WHI individuals. We selected transcripts with $p$ value <0.017 for follow-up analysis.

We validated that our approach to construct tPRS is robust. We compared a few polygenic prediction models developed using bulk RNA-seq in monocytes. First, the prediction model developed using prediXcan based on the MESA dataset[80,81], with weights provided in the predictDB database (http://www.predictdb.org/). Second, our approach above using genome-wide SNPs (including *trans*-eQTLs), and third, a similar clump and threshold approach as above limited to *cis*-eQTLs defined as SNPs within 1 Mbp of the start and end position of the transcript (the definition used by prediXcan). We focused on monocytes for this comparison because prediXcan models were only published based on monocytes.

**Using tPRS to identify reverse association between gene expression and SDB traits**. We constructed generalized tPRS in HCHS/SOL. We used HCHS/SOL genotypes imputed to the TOPMed freeze5b reference panel. Prior to tPRS construction, we filtered SNPs with imputation quality <0.8, minor allele frequency <5%,

missingness rate >0.01. As illustrated in panel b of Fig. 3, We identified potential reverse causation, where gene expression alters SDB, by using the tPRS constructed in HCHS/SOL as instrumental variables (IVs) and testing their association with their respective SDB phenotypes in HCHS/SOL. We used logistic mixed models, executed with the GENESIS R package[79] version 2.16.1. Each tPRS served as the exposure, and the relevant SDB phenotype served as the outcome. To account for skewness of the SDB phenotypes, we used the two-stage procedure for rank-normalization[77]. Relatedness was modeled via a sparse kinship matrix, household sharing, and block unit sharing among HCHS/SOL individuals. All association analyses were adjusted for age, sex, study site, Hispanic/Latino background, the first 5 PCs of the genetic data, and log of the sampling weights used to sample HCHS/SOL individuals into the study. Further, association analyses were adjusted to BMI when the tPRS and the SDB phenotype pair corresponded to an association between an SDB phenotype and a gene transcript in a BMI-adjusted analysis in MESA. In the analyses, all tPRS were standardized to have mean 0 and variance 1 in the HCHS/SOL dataset, so that effect size estimates correspond to 1 standard deviation (SD) increase in tPRS. Because the tPRS represent a genetic proxy for gene expression, if a tPRS was found to be associated with a SDB phenotype ($p$ value <0.05), it provided evidence that the transcript contributed to the SDB phenotype, rather than vice versa. However, as illustrated in diagrams a.2 and a.3 in Fig. 3 for sleep–transcript association, bidirectional associations are also plausible.

**Associations between tPRS and metabolites**. Treating tPRS as genetic IVs for gene expression, we estimated associations between tPRS and all identified (named) metabolites with <25% missing values in HCHS/SOL. We used robust survey models implemented in the R survey package version 4.0[82], accounting for HCHS/SOL study design (probability sampling and clustering) and providing associations generalizable to the HCHS/SOL target population. For each metabolite, we first imputed observations with missing values of that metabolite with its minimum value observed in the sample, under the assumption that missing values are due to concentrations being below the detection limit, and then rank-normalized it across the sample. We used the same covariates as before: age, sex (validated based on chromosomal checks), study site, Hispanic background, and the first 5 PCs of the genetic data. Furthermore, we adjusted for BMI depending on the original association of the SDB phenotype and the transcript (BMI unadjusted or BMI adjusted). As before, tPRS associations were estimated per 1 SD of the tPRS. For each transcript, we corrected metabolite associations to account for FDR using the Benjamini–Hochberg (BH) procedure[75]. Associations were considered significant if the FDR $p$ value was <0.05.

**Association analyses of SDB traits with selected metabolites to verify a complete association chain**. To further validate a complete association "chain" as detailed in Fig. 3, we performed association analyses between the SDB phenotypes and metabolites identified in the tPRS analysis. Associations between SDB phenotypes and metabolites used a survey sampling approach to account for HCHS/SOL sampling design and obtain estimates generalizable to the HCHS/SOL target population. Thus, we used the survey R package[83] with each individual weighted by their sampling weights, and clustering accounted for when computing robust standard errors. Analyses were adjusted for age, sex, study site, Hispanic/Latino background. and BMI depending on the original detected SDB-transcript association (BMI unadjusted or BMI adjusted). If an SDB phenotype was associated with the metabolite ($p$ value <0.05), we interpreted this as validation of a SDB association with this metabolite via the transcript-level chain.

**Reporting summary**. Further information on research design is available in the Nature Portfolio Reporting Summary linked to this article.

## Data availability

MESA, HCHS/SOL and WHI data are available through application to dbGaP according to the study specific accessions. MESA phenotypes are available in: "phs000209"; WHI phenotypes: "phs000200"; and HCHS/SOL phenotypes: "phs000810". HCHS/SOL genotyping data: "phs000880". MESA and WHI RNA-seq data has been deposited and will become available through the TOPMed according to the study specific accessions; MESA: "phs001416"; WHI: "phs001237". HCHS/SOL metabolomics data are available via data use agreement with the HCHS/SOL Data Coordinating Center at the University of North Carolina at Chapel Hill, see collaborators website: https://sites.cscc.unc.edu/hchs/. Data needed to construct the tPRS are publicly available on the following GitHub and zenodo repositories https://github.com/nkurniansyah/SDB_Multi_Omics, https://doi.org/10.5281/zenodo.7320074. Complete summary statistics from SDB traits association analyses with RNA-seq across cell types are provided in the same zenodo repository. The data behind Fig. 1 is provided in Supplementary Data 16.

## Code availability

We provide developed scripts used to perform analyses described in the paper and code to construct the tPRS in the GitHub repository https://github.com/nkurniansyah/SDB_Multi_Omics, https://doi.org/10.5281/zenodo.7320074.

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

## Acknowledgements
The authors thank the staff and participants of HCHS/SOL, MESA, and WHI, for their important contributions. We gratefully acknowledge the investigators and participants who provided biological samples and data for TOPMed. This work was supported by the National Heart Lung and Blood Institute grants R35HL135818 to S.R., T32-HL007901 to D.W., and R21HL145425 and R01HL161012 to T.S. Molecular data for the Trans-Omics in Precision Medicine (TOPMed) program was supported by the National Heart, Lung and Blood Institute (NHLBI). Genome Sequencing for "NHLBI TOPMed: Multi-Ethnic Study of Atherosclerosis (MESA)" (phs001416.v1.p1) was performed at the Broad Institute Genomics Platform (HHSN268201500014C). RNA-Seq for "NHLBI TOPMed: Multi-Ethnic Study of Atherosclerosis (MESA)" (phs001416.v1.p1) was performed at the Northwest Genomics Center (HHSN268201600032I). Genome Sequencing for "NHLBI TOPMed: Women's Health Initiative (WHI)" (phs001237.v3.p1) was performed at the Broad Institute Genomics Platform (HHSN268201500014C). RNA-Seq for "NHLBI TOPMed: Women's Health Initiative (WHI)" (phs001237.v3.p1) was performed at the Broad Institute Genomics Platform (HHSN268201600034I). Core support including centralized genomic read mapping and genotype calling, along with variant quality metrics and filtering were provided by the TOPMed Informatics Research Center (3R01HL-117626-02S1; contract HHSN268201800002I). Core support including phenotype harmonization, data management, sample-identity QC, and general program coordination were provided by the TOPMed Data Coordinating Center (R01HL-120393; U01HL-120393; contract HHSN268201800001I). The MESA projects are conducted and supported by the National Heart, Lung, and Blood Institute (NHLBI) in collaboration with MESA investigators. Support for the Multi-Ethnic Study of Atherosclerosis (MESA) projects are conducted and supported by the National Heart, Lung, and Blood Institute (NHLBI) in collaboration with MESA investigators. Support for MESA is provided by contracts 75N92020D00001, HHSN268201500003I, N01-HC-95159, 75N92020D00005, N01-HC-95160, 75N92020D00002, N01-HC-95161, 75N92020D00003, N01-HC-95162, 75N92020D00006, N01-HC-95163, 75N92020D00004, N01-HC-95164, 75N92020D00007, N01-HC-95165, N01-HC-95166, N01-HC-95167, N01-HC-95168, N01-HC-95169, UL1-TR-000040, UL1-TR-001079, UL1-TR-001420, UL1TR001881, DK063491, and R01HL105756. The authors thank the other investigators, the staff, and the participants of the MESA study for their valuable contributions. A fill list of participating MESA investigators and institutes can be found at http://www.mesa-nhlbi.org. The WHI program is funded by the National Heart, Lung, and Blood Institute, National Institutes of Health, U.S. Department of Health and Human Services through contracts 75N92021D00001, 75N92021D00002, 75N92021D00003, 75N92021D00004, 75N92021D00005. The Hispanic Community Health Study/Study of Latinos is a collaborative study supported by contracts from the National Heart, Lung, and Blood Institute (NHLBI) to the University of North Carolina (HHSN268201300001I/N01-HC-65233), University of Miami (HHSN268201300004I/N01-HC-65234), Albert Einstein College of Medicine (HHSN268201300002I/N01-HC-65235), University of Illinois at Chicago (HHSN268201300003I/N01- HC-65236 Northwestern Univ), and San Diego State University (HHSN268201300005I/N01-HC-65237). The following Institutes/Centers/Offices have contributed to the HCHS/SOL through a transfer of funds to the NHLBI: National Institute on Minority Health and Health Disparities, National Institute on Deafness and Other Communication Disorders, National Institute of Dental and Craniofacial Research, National Institute of Diabetes and Digestive and Kidney Diseases, National Institute of Neurological Disorders and Stroke, NIH Institution-Office of Dietary Supplements. The Genetic Analysis Center at the University of Washington was supported by NHLBI and NIDCR contracts (HHSN268201300005C AM03 and MOD03). Support for metabolomics data was graciously provided by the JLH Foundation (Houston, Texas).

## Author contributions
T.S. supervised the research. N.K. and T.S. conceptualized and conducted the biostatistical and bioinformatics analyses. N.K., T.S., and D.A.W. interpreted the results and wrote and edited the manuscript. B.Y., J.D.S., C.K., S.S.R., and J.I.R. designed metabolomics and/or RNA-seq data collections in either HCHS/SOL, MESA, or WHI. J.C., M.D., P.C.Z., R.K., and S.R. designed and participated in the collection of sleep measures in either HCHS/SOL or MESA. Y.Z., B.Y., B.C., H.W., H.M.O.-B., A.P.R., A.R.R., J.D.S., J.C., M.D., P.C.Z., R.K., C.K., S.S.R., J.I.R., S.A.G., and S.R. read and approved the final manuscript.

## Competing interests
S.R. reports receiving consulting fees from Jazz Pharma, Apnimed Inc, and Elli Lilly. These consultations were not related to the present work. All other authors declare no competing interests.
