## [Peer Review File · Communications Biology]

Reviewers' comments:

Reviewer #1 (Remarks to the Author):

The authors tried to integrate multi-omics data to investigate the underlying biological mechanisms of multiple SDB susceptibility variants. Overall, several points of certain novelty were delivered in this study.

Major comments

1. In addition to body mass index (BMI), did the authors identify any other risk factors for SDB, such as cardiometabolic proposed in the manuscript background? Will the results change after adjustment for other factors?
2. The results are inconsistent with the pictures provided, such as DNAJA3 expression in T cells. In addition, there are multiple genes of significantly increased, and the authors should describe more about the gene functions in either the result or discussion section.
3. The data in Table S11 do not adequately support this outcome analysis.
4. The conclusions needed to be expanded. The authors are suggested to highlight important findings and include afterthought of this work.

Minor:

5. Sleep Disordered Breathing (SDB) are defined multiple times in the manuscript. Please make sure using the acronym across the manuscript except the first time.
6. Line 78, line 549: multi-omics
7. The figure 3 is a bit blurry. Use high-resolution ones.

Reviewer #2 (Remarks to the Author):

Sleep-disordered breathing (SDB) is a very common disorder. However, how SDB affects the molecular environment is still poorly understood. In this paper, the authors identified an "association chain" among SDB traits (S), transcript expression (T), metabolites (M), and tPRS constructed with GWAS summary statistics (G). Such association chain is built with a series of association tests (listed below). Particularly, tPRS is introduced by the authors as a powerful bridge. The data sources, as well as how the data is collected, are clearly stated in the paper. I only have some minor questions/comments for some of the association tests:

Association between Transcripts (T) and SDB measures (S)

1. How is the joint association analysis (across 3 SDB measures) done? For example, by Cauchy combination test (CCT)?
2. The multiple testing and FDR correction is done across all SDB measures for each transcript, or across all transcripts for each SDB measure, or both?
3. The above two questions then motivate this question: in Table 1, why there is only one (FDR) p-value for each gene? I am thinking there would be 4 p-values (3 for individual SDB measures and 1 for joint SDB measure) for each gene. Or the authors are just showing the significant one?

Association between tPRS (G) and Transcripts (T)

1. Why is BMI not used for adjustment here?
2. In line 283, please clarify what does "three tPRS" mean? Does it mean the three tPRS with three different p-value thresholds? If that so, how the authors determine the final tPRS from three candidate tPRSs, or how the authors decide the optimal choice of p-value cutoff?
3. Since the authors have a series of tPRSs (96 and 24), did the authors standardize them before pool them for further analysis?

Association between tPRS (G) and SDB measures (S): Clear

Association between tPRS (G) and Metabolites (M): Clear

Association between SDB measures (S) and Metabolites (M): Clear

Reviewer #1 (Remarks to the Author):

The authors tried to integrate multi-omics data to investigate the underlying biological mechanisms of multiple SDB susceptibility variants. Overall, several points of certain novelty were delivered in this study.

Major comments

1. In addition to body mass index (BMI), did the authors identify any other risk factors for SDB, such as cardiometabolic proposed in the manuscript background? Will the results change after adjustment for other factors?

Response: Thank you for this excellent comment. We now added such an analysis. We recently published a manuscript that used Mendelian Randomization analysis to identify causal associations between SDB and other a range of other cardiometabolic and glycemic traits. We now used the phenotypes suggested as causal to SDB in a secondary analysis, adjusting to them. We also added alcohol use to the list of variables, as it is another risk factor, and because ethanol is an inhibitor of *P2RX4*. We first report this in the methods section (page 19):

“In secondary analyses we assessed the effect of adjusting for additional available phenotypes that prior work suggested as causal to SDB (74), and for alcohol use. Specifically, we applied the Olivia pipeline to estimate the same transcript-SDB phenotype associations using the same regression adjustment, while also including one of the additional risk factors as a covariate: pulse pressure, type 2 diabetes, waist-to-hip ratio, hemoglobin A1c, and alcohol use. An additional analysis included these five covariates together. We studied the association results for all transcripts with FDR p-value<0.1 in the main analyses once adjusting to these covariates. Description of these measures is provided in the Supplementary Materials.”

Next, we report the results in the results section (page 5):

“Supplementary Table 4, 5, 6, and 7 further report results from secondary analyses adjusting all associations with FDR p-value<0.1 in the primary analysis to cardiometabolic causal risk factors of OSA, including pulse pressure, type 2 diabetes, waist-to-hip ratio, hemoglobin A1c, as well as to alcohol use. Adjustments were performed for each phenotype separately, and jointly. Throughout, association effect estimates remained similar to those from the primary analysis.”

2. The results are inconsistent with the pictures provided, such as DNAJA3 expression in T cells. In addition, there are multiple genes of significantly increased, and the authors should describe more about the gene functions in either the result or discussion section.

Response: Thank you for carefully reviewing and identifying errors. We updated the text to fix this error and added description of other genes. For example, in the results section, subsection “SDB phenotypes for oxyhemoglobin saturation and AHI are linked to tissue-specific changes in the transcriptome”, on page 6, we updated the text regarding differences in gene expression across tissues:

“There are a few striking differences in gene expression, particularly the increased expression of *FAM106A*, *TMC3-AS1*, *SERPINE2*, *LA16c-312E8.4*, and *DUX4L27* in T-cells in association with better SDB measures (lower AHI, higher MinO2 and AvgO2) compared to monocytes and PBMCs in which the

corresponding expressions tended to decrease, whereas expressions of *EMP1*, *SIAE*, *PDGFC*, and *LHFPL2* were decreased across tissues in improved SDB phenotypes. To further investigate the overall patterns in gene expression in relation to tissue type and SDB traits, a heatmap of the Spearman correlation of the log-fold expression estimates of SDB phenotypes was plotted in Figure 3. Within cell types, the SDB traits AHI and MinO2 had the highest correlation for gene expression (Spearman R^2 between 0.91 to 0.97), whereas AvgO2 associations had lower correlations with AHI and MinO2 associations, especially in monocytes.”

We also expanded the write up about various genes in the discussion. For example, on page 9 , we edited and expanded the write up about *EMP1* and added text about *SIAE* and *LHFPL2*:

“In monocytes, increased expression of *EMP1*, *SIAE*, and *LHFPL2* was associated with lower MinO2, suggesting that expression of these genes increases as oxygen levels decrease. In line with these findings, prior studies have shown that *EMP1* expression increases during sleep loss and during hypoxia in cancer tissues (32,33). While the functions of the esterase *SIAE* encodes and the transmembrane protein *LHFPL2* encodes are unclear, increased expression of each is a poor prognostic marker for cancer; higher expression of *SIAE* is linked to poorer prognosis for patients with multiple myeloma and higher expression of *LHFPL2* is linked to poorer prognosis for patients with liver cancer (34,35). These cancer-related markers may be relating to SDB phenotypes due to the overlap between hypoxia and the hypoxic tumor microenvironment (36).”

Added information and interpretation about *TMC3-AS1* on page 10:

“Similar to *FAM106A* and *SERPINE2*, *TMC3-AS1* expression in T-cell increased as AHI increased; because *TMC3-AS1* encodes a lncRNA that may inhibit the anti-inflammatory cytokine IL-10 (45), increased expression could result in higher IL-10 levels, possibly as a compensatory response to greater AHI.”

Added discussion of the potential role of ethanol, as an inhibitor of *P2RX4* on page 11:

“However, ethanol is an inhibitor of *P2RX4*, and *P2RX4* has been associated with alcohol intake. Because alcohol consumption is a risk factor for SDB and can promote airway collapse(54–56), in a secondary analysis we adjusted for self-reported alcohol consumption in the MESA RNA-seq analysis. The results did not substantially change, suggesting that alcohol use is not driving this association; however, residual confounding by alcohol use is still possible.”

And expanded the discussion regarding results related to *CTD-2366F13.1* (page 11):

“While *CTD-2366F13.1* (*MOCS2-DT*) PBMCs tPRS was not associated with SDB traits in HCHS/SOL, levels of three of the four metabolites positively associated with *CTD-2366F13.1* tPRS were also negatively associated with AvgO2 (i.e., increased metabolite levels with reduced AvgO2): linoleoyl-arachidonoyl-glycerol (18:2/20:4), palmitoleoyl-linoleoyl-glycerol (16:1/18:2), and 1-stearoyl-2-arachidonoyl-GPE (18:0/20:4). The lack of observed association between the *CTD-2366F13.1* tPRS and AvgO2 suggests that it is likely that AvgO2 may cause expression changes in the gene, rather than the genetically-determined gene expression causes AvgO2 (Figure 2). Therefore, lower AvgO2 may result in increased concentrations of these metabolites. linoleoyl-arachidonoyl-glycerol (18:2/20:4), has previously been positively associated with serum levels of the antioxidant alpha-tocopherol, also known as Vitamin E (57). Levels of the

component 2-Arachidonoylglycerol, an agonist of the CB1 and CB2 cannabinoid receptors, are increased in the brain during ischemia (58) and in macrophages in response to oxidative stress (59). Palmitoleoyl-linoleoyl-glycerol (16:1/18:2), is a palmitoleic acid derivative that may be a marker of blood sugar regulation; it is also commonly used in baked goods (60). Increased levels have previously been linked to in utero exposure to gestational diabetes (60). Given the relationship between SDB and cardiometabolic disease, it is possible that these metabolites are associated with AvgO2 because of their links to the immune system and glycemic regulation. In fact, AvgO2 associations with these metabolites became null in a BMI-adjusted analysis.”

We also added a few other, smaller clarifications and improvements in other places (see tracked changes).

3. The data in Table S11 do not adequately support this outcome analysis.

Response: The reviewer is correct. Thank you for your careful review, we identified an important mistake that we made and were able to fix it. In detail: Table S11 reports the association of generalized tPRS with SDB phenotypes in HCHS/SOL. In the previous version, the association of the tPRS for *P2RX4* with AvgO2 was positive (higher tPRS increasing AvgO2). The association of the transcript expression with AvgO2 in MESA was negative: higher AvgO2 was associated with reduced expression of *P2RX4*. We carefully reviewed our code and the genetic files and found that we made an error when analyzing HCHS/SOL data, where we switched the effect and the other allele. Now the direction of associations of the tPRS for *P2RX4* with AvgO2 in HCHS/SOL matches the direction of association of AvgO2 with *P2RX4* expression in MESA. This error, unfortunately, led to additional errors, including in the direction of associations of the tPRS with metabolites. We fixed these errors and carefully updated the text.

For example, we updated the text in the results section, subsection “Evidence of causal association between transcripts and SDB phenotypes”, page 8:

“...Among them, the strongest association was of the tPRS for *P2RX4* (Purinergic Receptor *P2X 4*) in PBMCs in its association with AvgO2; **one standard deviation (SD) increase in the tPRS was associated with a 1.9% decrease in AvgO2. Additionally, higher tPRS for *TUBB6* (Tubulin Beta 6 Class V) in monocytes was associated with lower MinO2. These directions of associations matched those observed in association analysis of the SDB phenotype and transcript expression in MESA. However, higher tPRS for *SEC14L2* (SEC14 Like Lipid Binding 2) in T-cells was associated with higher AHI, but this direction of association did not match that of the estimated AHI-transcript association in MESA. After BMI adjustment, only *P2RX4* in PBMCs tPRS remained associated with AvgO2 (p-value <0.05), as shown in Supplementary Table 11.”**

In addition, we went through the manuscript and updated various statements about “positive/negative associations” to be more explicit, i.e., of the form: “higher expression levels were associated with lower AvgO2”. With this, we reviewed every statement while looking at the tables, and we think that report of findings is more accurate now (see tracked changes).

4. The conclusions needed to be expanded. The authors are suggested to highlight important findings and include afterthought of this work.

Response: Thank you for this suggestion. We updated the conclusion paragraph on page 14 (new text in bold):

“In summary, we examined multiple levels of biological information to investigate signaling mechanisms underlying SDB traits to better understand drivers of morbidity in SDB. Our results highlight differential gene expression by circulating leukocyte populations in relation to multiple SDB traits related to hypoxia, neurotransmission, and thrombolytic activity. Analyses with validated tPRS in independent cohorts support a mechanistic role for *P2XR4* purinergic signaling in SDB, a gene known to influence cardiac function, which is relevant to SDB as both a risk factor and an outcome. **While further research is necessary to confirm these findings, they suggest that *P2XR4* signaling may alter oxygen levels during sleep. In the future, we hope that more data will become available with more granular cell-specific transcriptomics, to better understand cell-specific responses involved in SDB, as well as their validation. Large complementary genetic datasets with unbiased genetic associations of SDB, gene transcripts, and metabolomics, will further facilitate causal inference via Mendelian randomization analysis.** Overall, we applied robust methods to integrate multi-omics data and SDB data to discover mechanisms underlying multiple SDB traits. Our multi-dimensional approach using large population cohorts is a promising approach to unravel biological underpinnings of complex human disorders.”

Minor:

5. Sleep Disordered Breathing (SDB) are defined multiple times in the manuscript. Please make sure using the acronym across the manuscript except the first time.

Response: Done.

6. Line 78, line 549: multi-omics

Response: Done.

7. The figure 3 is a bit blurry. Use high-resolution ones.

Response: Done. We are also uploading the figure files separately (in addition to embedding them in the text), hopefully this will help.

Reviewer #2 (Remarks to the Author):

Sleep-disordered breathing (SDB) is a very common disorder. However, how SDB affects the molecular environment is still poorly understood. In this paper, the authors identified an "association chain" among SDB traits (S), transcript expression (T), metabolites (M), and tPRS constructed with GWAS summary statistics (G). Such association chain is built with a series of association tests (listed below). Particularly, tPRS is introduced by the authors as a powerful bridge. The data sources, as well as how the data is collected, are clearly stated in the paper.

Response: Thank you for your thorough review and for the helpful suggestions and comments! We hope that the methods are clearer after addressing your comments.

I only have some minor questions/comments for some of the association tests:

Association between Transcripts (T) and SDB measures (S)

1. How is the joint association analysis (across 3 SDB measures) done? For example, by Cauchy combination test (CCT)?

Response: The joint association analysis was performed by using the three SDB measures together in a single regression model. This approach was previously described in PMID: 34015820. We clarified this in the methods section, page 19:

“In joint analysis, the three SDB phenotypes were modeled as three exposures in the same model.”

And clarified the testing procedure, in page 19:

“However, for joint SDB phenotypes analysis we used the multi-variate Wald test, without further empirical p-values, because the permutation procedure only works when using a single exposure.”

2. The multiple testing and FDR correction is done across all SDB measures for each transcript, or across all transcripts for each SDB measure, or both?

Response: The multiple testing FDR correction was applied across all transcripts for each SDB measures. We clarify this in the methods section, page 19 (new text in bold):

“Finally, we accounted for multiple testing by applying False-Discovery Rate (FDR) correction to each set of gene-based p-values corresponding to a single SDB phenotype, or the joint association, using the Benjamini-Hochberg (BH) procedure (73).”

3. The above two questions then motivate this question: in Table 1, why there is only one (FDR) p-value for each gene? I am thinking there would be 4 p-values (3 for individual SDB measures and 1 for joint SDB measure) for each gene. Or the authors are just showing the significant one?

Response: Yes, we are only showing the significant ones, as the reviewer observed, the reason is the massive number of results (many columns/rows that would potentially be displayed). To clarify, we added in the caption of Table 1 (new word in bold):

“The table provides significant results from analyses of unadjusted (n=8 transcripts) and adjusted (n=4 transcripts) for BMI.”

Supplemental Figures 3 and 4 visualize all results (albeit without effect size and p-values), and we provide complete results on zenodo (<https://doi.org/10.5281/zenodo.7320074>), as we now describe in the data availability statement (page 31):

“Data needed to construct the tPRS are publicly available on the repository https://github.com/nkurniansyah/SDB_Multi_Omics, <https://doi.org/10.5281/zenodo.7320074>. Complete summary statistics from SDB traits association analyses with RNAseq across cell types are provided in the same zenodo repository”.

Association between tPRS (G) and Transcripts (T)

1. Why is BMI not used for adjustment here?

Response: It was adjusted for BMI, we apologize for miscommunicating. We clarified this in the methods section, page 22, new text in bold:

“All association analyses were adjusted for age, sex, study site, Hispanic/Latino background, the first 5 PCs of the genetic data, and log of the sampling weights used to sample HCHS/SOL individuals into the study. Further, association analyses were adjusted to BMI when the tPRS and the SDB phenotype pair corresponded to an association between an SDB phenotype and a gene transcript in a BMI-adjusted analysis in MESA.”

2. In line 283, please clarify what does “three tPRS” mean? Does it mean the three tPRS with three different p-value thresholds? If that so, how the authors determine the final tPRS from three candidate tPRSs, or how the authors decide the optimal choice of p-value cutoff?

Response: Indeed, we considered three p-value threshold. This is written in the methods section and now further clarified, page 20, and includes an explanation of how we decided on the optimal p-value threshold – by “validation” in the independent WHI dataset (new text in bold):

“We applied clump and threshold methodology implemented in PRSice2 v2.3.1.e (76) using clumping parameters $R^2=0.1$, distance of 250Kb, and three p-value thresholds: genome-wide significance (5×10^{-8}), and levels of evidence considered “suggestive” (10^{-7} , 10^{-6}). For each transcript, we constructed the three tPRS in WHI. A tPRS with the smallest p-value in association with the transcript in WHI, and also having $p\text{-value} < 0.05/3 = 0.017$, was selected and considered validated.”

3. Since the authors have a series of tPRSs (96 and 24), did the authors standardize them before pool them for further analysis?

Response: We did not pool the tPRSs, i.e., we used them in association analyses individually. The tPRS were standardized in all association analyses, so that results are described in terms of 1SD increase in the tPRS. We clarified this on page 22:

“In analyses, all tPRS were standardized to have mean 0 and variance 1 in the HCHS/SOL dataset, so that effect size estimates correspond to 1 standard deviation (SD) increase in the relevant tPRS.”

We also clarified this in page 23 (when describing the analysis of tPRS associations with metabolites):

“As before, tPRS associations were estimated per 1 SD of the tPRS.”

REVIEWERS' COMMENTS:

Reviewer #1 (Remarks to the Author):

All my comments have been addressed. No more comment.

Reviewer #2 (Remarks to the Author):

The authors have addressed all my questions. The way they performed association tests between transcripts and SDB measures, and between tPRS and transcripts are now clearly clarified (including multiple testing approaches and the corresponding results). I don't have any further comments for the manuscript.